# Changes in Anterior Chamber Angle and Choroidal Thickness in Patients with Primary Angle-Closure Glaucoma after Phaco-Goniosynechialysis

**DOI:** 10.3390/jcm12020406

**Published:** 2023-01-04

**Authors:** Surong Luo, Guomei Yuan, Chenwei Zhao, Jiang Shen, Li Zhang, Man Luo, Wei Chen

**Affiliations:** Department of Ophthalmology, Shaoxing People’s Hospital, Zhejiang University, Shaoxing 312000, China

**Keywords:** primary angle closure glaucoma, goniosynechialysis, anterior chamber angle width, choroidal thickness

## Abstract

We aimed to observe changes in angle width and choroidal thickness (CT) before and after phacoemulsification intraocular lens implantation (PEI) combined with goniosynechialysis (GSL) in patients with primary angle-closure glaucoma (PACG) complicated by cataracts. This prospective cohort study included 60 patients with PACG complicated by cataracts from the Department of Ophthalmology of Shaoxing People’s Hospital, China. Patients underwent PEI combined with GSL (PEI-GSL group) or laser peripheral iridotomy (LPI) followed by PEI (PEI group). Intraocular pressure (IOP) and endothelial counts were significantly decreased in both groups after surgery, while best-corrected visual acuity and central anterior chamber depth were significantly increased. However, there were no significant differences between the two groups. The opening degrees of room corners at 12, 3, 6, and 9 o’clock were recorded as AA12, AA3, AA6, and AA9, respectively. Anterior chamber depth was significantly increased postoperatively compared to preoperatively in both groups, with no significant between-group differences (all *p*s > 0.05). At 1, 3, and 6 months postoperatively, the width at AA12, AA3, and AA9 points was higher in the PEI-GSL group than in the PEI group (all *p*s < 0.05). Significant between-group differences at AA6 were observed preoperatively (*p* = 0.023) and at 1 (*p* = 0.027) and 3 (*p* = 0.033) months postoperatively but not at 6 months postoperatively (*p* = 0.055). CT was smaller postoperatively than preoperatively (all *p*s < 0.001). The present study suggests that patients with PACG who underwent PEI with or without GSL had reduced IOP and CT after surgery.

## 1. Introduction

Glaucoma, a group of heterogeneous optic neuropathies, is the second most common cause of blindness worldwide and is the most common cause of irreversible blindness. It disproportionately affects women and Asians. Primary angle-closure glaucoma (PACG) is Asia’s most common type of glaucoma [1]. PACG is more common in elderly individuals and often presents complications, such as senile cataracts and the aggravation of the pupillary block due to lens characteristics [2]. According to the 2019 expert consensus on the diagnosis and treatment of PACG [3], phacoemulsification and intraocular lens implantation are recommended as the first choices for these patients, and goniosynechialysis (GSL) is performed under a gonioscope [4]. However, the optimal postoperative approach for changing anterior chamber depth and angle and choroidal thickness (CT) in PACG patients with senile cataracts has not been determined. Further, it remains unclear whether there are differences in anterior chamber depth, angle width, and CT between patients undergoing combined GSL and those undergoing phacoemulsification and intraocular lens implantation after laser peripheral iridotomy (LPI). Indeed, the exact role of CT in glaucoma remains undetermined [5]. Accordingly, the aim of this study was to observe changes in angle width and CT in patients with PACG complicated by cataracts before and after surgery. We compared changes in chamber angle width and CT postoperatively between patients who underwent phacoemulsification and intraocular lens implantation combined with GSL (PEI-GSL group) and those who underwent LPI followed by phacoemulsification and intraocular lens implantation (PEI group). Furthermore, we also investigated changes in CT postoperatively compared with preoperatively in these two groups.

## 2. Materials and Methods

### 2.1. Patients

A total of 60 patients diagnosed with PACG with cataracts at the Department of Ophthalmology of Shaoxing People’s Hospital between November 2020 and November 2021 were selected. According to the admission order, phacoemulsification and intraocular lens implantation combined with GSL were performed in the PEI-GSL group. Patients in the PEI group were treated with LPI, followed by phacoemulsification and intraocular lens implantation. All participants provided informed consent prior to participation. This study was conducted in compliance with the Declaration of Helsinki and was approved by the hospital’s Ethics Committee (No: 2019-K-Y-197-01).

Inclusion criteria were the diagnosis of PACG and meeting cataract requirements. PACG was diagnosed by the presence of the following characteristics: (1) anatomical features of angle-closure glaucoma, shallow central and peripheral anterior chambers; (2) acute or chronic anterior chamber angle closure, anterior chamber angle adhesion closure range ≤ 180°; (3) intraocular pressure (IOP) below 30 mmHg (1 mmHg = 0.133 kPa) after drug control; and (4) different degrees of visual field defects shown by visual field examination. A diagnosis of cataracts had the following requirements: (1) cortical or nuclear opacity of the lens and (2) corrected visual acuity of ≤0.5.

### 2.2. Ophthalmic Examinations

All patients underwent the following ophthalmic examinations: best corrected visual acuity (BCVA) testing, IOP measurement, slit-lamp microscope examination, anterior segment examination, corneal endothelial count, gonioscopy, Humphrey field testing, ultrasound biomicroscope (UBM) examination, and enhanced depth imaging spectral-domain optical coherence tomography (EDI-OCT). BCVA, IOP, corneal endothelial count, and CT scores were recorded preoperatively and at 1 week, 1 month, 3 months, and 6 months postoperatively for all PACG patients with cataracts. The central anterior chamber depth (CACD) and anterior chamber angle width (12, 3, 6, and 9 o’clock) of all PACG patients with cataracts were recorded preoperatively and at 1, 3, and 6 months postoperatively. The same experienced examiner performed all examinations.

#### 2.2.1. BCVA

A standard logarithmic visual acuity chart was used to examine monocular BCVA at a distance of 5 m. The visual acuity values were documented using LogMAR recording.

#### 2.2.2. IOP

IOP was measured using a Goldmann flattening tonometer (Haag-Streit, AT900, Switzerland). A mean IOP value was recorded after three repeated measurements during the same time (8–10 am daily).

#### 2.2.3. Corneal Endothelium

A PERSEUS model corneal endothelium microscope (CSO, Italy) was used to automatically analyze corneal endothelium data after proper head positioning.

#### 2.2.4. Gonioscopy

The anterior chamber angle of 360° was examined using gonioscopy, which consisted of static and dynamic examinations. Patients were instructed to gaze naturally at the front of the anterior chamber. A short narrow beam was used to avoid applying pressure during gonioscopy while avoiding prism tilt. The 360° anterior chamber angle was observed directly through the reflection and refraction of mirrored light. If the patient’s anterior chamber angle was observed to be narrow, dynamic anterior chamber angle microscopy was performed. Patients were instructed to turn their eyes toward the mirror and change their eyeball position. The pressure was applied to expose the anterior chamber angle depth fully.

#### 2.2.5. Humphrey Vision

All participants were corrected for refractive error and presbyopia according to visual proximity requirements using the Humphrey 750i visual field meter (24-2 mode; Carl Zeiss, USA) under the threshold detection program and SITa-FAST strategy, with a background light of 31.5 asb. The visual marker was the Goldmann III white visual marker with physiological blind spot-fixed vision monitoring. The morphological characteristics of visual field damage presented in the pattern deviation map (including a probability map and numerical map) in the printed report of single-field analyses were considered the main reference, including paracentric scotoma, nasal ladder, or more severe visual field morphological defects.

#### 2.2.6. CACD and Anterior Chamber Angle Width

The CACD and anterior chamber angle width at different quadrant points (12, 3, 6, and 9 o’clock) were measured using a UBM. CACD was defined as the tangent distance from the center of the posterior corneal surface to the anterior surface of the lens. The anterior chamber angle width was defined as the open angle (AA). A circle with a radius of 500 μm was made using the scleral process as the apex of the chamber angle and the center of the circle. The angle formed from the intersection of the circle with the endothelial surface of the cornea and the anterior surface of the iris, as the two ends of the chamber angle, was defined as the AA. The opening degrees of room corners at 12, 3, 6, and 9 o’clock were recorded as AA12, AA3, AA6, and AA9, respectively.

#### 2.2.7. CT

EDI-OCT software (Heidelberg, Germany) was used to measure CT. Patients were instructed to place their jaw on the jaw rest and gaze at the instrument’s internal fixation cursor or auxiliary external fixation light. The macular was scanned through the horizontal line of the fovea. Each OCT image included 100 images that were automatically tracked, superimposed, and enhanced. CT was measured based on the vertical distance between the outer retinal pigment epithelium and the inner boundary of the sclera. Subfoveal (SF), nasal 1 (N1, 0.5 mm from the SF), nasal 2 (N2, 1.5 mm from the SF), nasal 3 (N3, 3.0 mm from the SF), temporal 1 (T1, 0.5 mm from the SF), temporal 2 (T2, 1.5 mm from the SF), and temporal 3 (T3, 3.0 mm from the SF) were measured at seven points.

### 2.3. Surgical Method

Local and systemic medications (pilocarpine 4×/d, brinzolamide 2×/d, carteolol 2×/d, brimonidine 3×/d, and mannitol 100 mL 2×/d) were applied preoperatively to reduce IOP. Patients in the PEI group underwent LPI, and all patients’ IOP values decreased to less than 30 mmHg. Compound tropicamide eye drops were applied for pupil dilation. The operative eye was anesthetized with 5% hydrochloric acid Erkine eye drops. After local disinfection was completed and the disinfectant towels were spread, eyelids were opened using an opening device. A temporal corneal incision was made with a 1.5 mm bayonet. A small amount of aqueous humor was released to reduce IOP, and an appropriate amount of sodium hyaluronate was injected through the lateral incision. Subsequently, a clear corneal microincision was made with a 1.8 mm bayonet. Annular capsulorhexis was performed with capsulorhexis forceps, followed by water separation and phacoemulsification to aspirate the lens nucleus and cortex. A posterior chamber intraocular lens was implanted after polishing the posterior capsule. Carbachol was injected to narrow the pupil. In the PEI-GSL group, the anterior chamber angle was obtusely separated by slowly injecting sodium hyaluronate into the anterior chamber angle for 360° angle separation. After satisfactory separation through gonioscopy, the sodium hyaluronate was removed. The anterior chamber was then rinsed, and a watertight incision was made. Finally, a subconjunctival injection of dexamethasone (1.0 mg) was administered.

### 2.4. Statistical Analysis

This prospective cohort study analyzed data using Statistical Package for Social Sciences software version 25 (IBM Corp, Armonk, NY, USA). Data were expressed as means and standard deviations, ranges, and/or percentages, as appropriate. The normality of the data distributions was assessed using the Shapiro–Wilk test. The repeated measures analysis of variance (ANOVA) was used to compare the data at different time points for each group. A test for sphericity was first performed for each dataset. *p* > 0.05 indicated the lack of a correlation between the data at different time points and that the measured data met the Huynh–Feldt condition. If the data were also normally distributed, the data were analyzed using a one-way ANOVA. *p* < 0.05 for the sphericity test indicated a correlation between the data at different time points. A repeated measures ANOVA was performed, along with the least significant difference (LSD) method, to compare the data at different time points. An independent samples *t*-test was used to compare the data between the two groups. Levene’s variance test was performed to evaluate the homogeneity of variance; if the significance level was >0.05, the variance was uniform (see the result of “assumed isovariance” in the first line); if the significance was <0.05, the variance was non-uniform (see the result of “equivariance was not assumed” in the second line). Values were considered statistically significant at *p* < 0.05. All reported *p*-values were two-sided. Patients were divided into groups through single blinding (only the study participants were blinded).

## 3. Results

### 3.1. Basic Characteristics of Patients

In total, three patients were lost to postoperative follow-up in the PEI-GSL group because sequential follow-up data could not be obtained; therefore, they were excluded from the trial. There were no significant differences in age, sex, or the ratio of acute angle-closure glaucoma/chronic angle-closure glaucoma (CACG) between the PEI-GSL and PEI groups. Basic information is presented in Table 1.

### 3.2. BCVA, IOP, and Corneal Endothelium

BCVA showed improvement at various time points based on LSD analyses (all *p*s < 0.001) (Table 2). Significant differences in BCVA were observed at different time points pre- and postoperatively in the PEI-GSL (F_PEI-GSL_ = 36.837, *p* < 0.001) and PEI (F_PEI_ = 26.534, *p* < 0.001) groups. 

Significant differences in IOP were observed at different time points pre- and postoperatively in the PEI-GSL (F_PEI-GSL_ = 4.580, *p* < 0.001) and PEI (F_PEI_ = 10.075, *p* < 0.001) groups. LSD analyses revealed that IOP was lower at different time points postoperatively than preoperatively (all *p*s < 0.05). In the PEI-GSL group, IOP was significantly lower at 1, 3, and 6 months postoperatively than at 1 week postoperatively (all *p*s < 0.05). No significant differences in IOP were noted at different postoperative time points in the PEI group (all *p*s > 0.05), and no significant difference in the preoperative IOP was observed between the two groups (t = 0.533, *p* = 0.598). IOP was lower in the PEI-GSL group than in the PEI group at 6 months postoperatively (t = −2.629, *p* = 0.011) (Table 2).

A comparison of the preoperative corneal endothelial counts between the PEI-GSL group and the PEI group showed no statistical significance (t = 0.022, *p* = 0.982). A significant difference was observed in corneal endothelial counts at different time points pre- and postoperatively in the PEI-GSL (F_PEI-GSL_ = 12.903, *p* < 0.001) and PEI (F_PEI_ = 17.011, *p* < 0.001) groups. LSD analyses revealed that the corneal endothelial counts of the two groups were decreased at various postoperative time points (all *p*s < 0.05). Corneal endothelial counts were significantly higher at 6 months postoperatively than at 1 week, 1 month, and 3 months postoperatively in the PEI group (all *p*s < 0.05) (Table 3).

### 3.3. CACD and AA

There was no significant difference in preoperative CACD between the PEI-GSL group and the PEI group. A significant difference was observed in CACD at different time points pre- and postoperatively in the PEI-GSL (F_PEI-GSL_ = 123.762, *p* < 0.001) and PEI (F_PEI_ = 120.247, *p* < 0.001) groups. LSD analyses revealed significant increases in CACD in the PEI-GSL group at 1, 3, and 6 months postoperatively (all *p*s < 0.05). In the PEI group, CACD increased sequentially at 1 and 3 months postoperatively (all *p*s < 0.05). No significant difference in CACD was noted between 3 and 6 months postoperatively in the PEI group (*p* > 0.05) (Table 4).

The preoperative AA12, AA3, AA6, and AA9 values for the PEI-GSL group were 4.30 ± 7.08°, 11.92 ± 8.52°, 6.95 ± 8.52°, and 9.87 ± 9.63°, respectively. Significant differences in AA were noted at various time points pre- and postoperatively (F_AA12_ = 50.496, F_AA3_ = 56.562, F_AA6_ = 74.745, and F_AA9_ = 48.446; all *p*s < 0.001). In the PEI group, preoperative AA12, AA3, AA6, and AA9 were 8.17 ± 9.82°, 13.70 ± 9.41°, 12.77 ± 10.12°, and 14.41 ± 9.79°, respectively. Significant differences in AA were observed at different time points pre- and postoperatively (F_AA12_ = 7.760, *p* = 0.001; F_AA3_ = 11.057, F_AA6_ = 8.581, and F_AA9_ = 15.149; all *p*s < 0.001).

AA was compared between the PEI-GSL and PEI groups at different time points pre- and postoperatively. Preoperative AA6 was narrower in the PEI-GSL group than in the PEI group (t_AA6_ = −2.335, *p* = 0.023), and there was no significant difference in AA6 between the PEI-GSL and PEI groups at 6 months postoperatively (t_AA6_ = 1.957, *p* = 0.055). There were no significant between-group differences in preoperative AA12 (t_AA12_ = −1.721), AA3 (t_AA3_ = −0.747), or AA9 (t_AA9_ = −1.762) (all *p*s > 0.05). AA12, AA3, and AA9 were larger in the PEI-GSL group than in the PEI group at 1, 3, and 6 months postoperatively (all *p*s < 0.05) (Table 5).

### 3.4. CT

Comparisons between the PEI-GSL and PEI groups of preoperative CT at seven points revealed that the CT of SF, N1, N2, T1, and T2 was thinner in the PEI-GSL group than in the PEI group (all *p*s < 0.05). No significant between-group differences in CT at N3 (t_N3_ = −1.383, *p* = 0.175) or T3 (t_T3_ = −1.948, *p* = 0.060) were observed (Table 6). In both groups, CT was the thickest in the SF position, and it was significantly thicker on the temporal side than on the nasal side (all *p*s < 0.05).

In the PEI-GSL group, the spherical test results of CT preoperatively and at 1 week, 1 month, 3 months, and 6 months postoperatively revealed a correlation between CT at different time points (*p* < 0.001). Therefore, a repeated measures ANOVA was performed. The analysis revealed significant differences in CT between the five pre- and postoperative time points in the PEI-GSL group (all *p*s < 0.001) (Table 7). Pairwise comparisons, completed using the LSD method, revealed that the CT of the PEI-GSL group was significantly thinner at each time point postoperatively than preoperatively (all *p*s < 0.01). The CT of the PEI-GSL group was significantly thinner at 6 months postoperatively than at 1 week, 1 month, and 3 months postoperatively (all *p*s < 0.05). No significant differences in CT were observed at 1 week, 1 month, or 3 months postoperatively (all *p*s > 0.05).

In the PEI group, the spherical test results of CT preoperatively and at 1 week, 1 month, 3 months, and 6 months postoperatively revealed a correlation between CT at different time points (*p* < 0.001). Therefore, a repeated measures ANOVA was performed. The analysis revealed significant differences in CT between the five pre- and postoperative time points in the PEI group (all *p*s < 0.001) (Table 8). Pairwise comparisons, completed using the LSD method, revealed that the CT of the PEI group was significantly thinner at each time point postoperatively than preoperatively (all *p*s < 0.01). CT at N1, N2, N3, SF, T1, and T2 was significantly thinner at 6 months postoperatively than at 1 week, 1 month, and 3 months postoperatively (all *p*s < 0.05). At T3, no significant difference in CT was observed between 3 and 6 months postoperatively (*p* = 0.652).

Comparisons of postoperative CT at seven points between the PEI-GSL and PEI groups revealed that: (1) at N3, there was no statistical difference in CT between the two groups at 1 week, 1 month, 3 months, or 6 months postoperatively (all *p*s > 0.05); (2) at T3, there was no statistical difference in CT at 3 months postoperatively (*p* = 0.062), but the comparison of CT at 1 week, 1 month, and 6 months postoperatively suggested that the CT of the PEI-GSL group was thinner than that of the PEI group (all *p*s < 0.05); (3) at N2, N1, SF, T1, and T2, the CT of the PEI-GSL group was thinner than that of the PEI group (all *p*s < 0.05).

## 4. Discussion

PACG is characterized by the acute or chronic elevation of IOP caused by primary angle closure, with or without glaucomatous optic disc changes and visual field damage [6]. The pathogenesis of angle-closure glaucoma not only involves changes in anatomical factors but also can be explained by genome-related studies [5,7]. Anatomical mechanisms can be divided into the pupillary block, non-pupillary block, and mixed mechanism types. The expert consensus on the diagnosis and treatment of PACG in China (2019) classified this condition according to the mechanisms of chamber angle closure [3], including traditional pupillary block mechanisms and non-pupillary block mechanisms, such as plateau iris, anterior ciliary body position, abnormal lens position, and choroidal expansion. Wang et al. [8] reported that mixed mechanisms caused 54.8% of the angle-closure glaucoma cases in China; pupillary and non-pupillary block types accounted for approximately 38.1% and 7.8%, respectively. The pupil and relative lens position are referred to as “physiologic pupillary block”. A pathological pupillary block may occur if the iris sphincter and anterior lens capsules are in close contact. For example, aqueous humor from the posterior chamber through the pupil to the anterior chamber may increase resistance and pressure behind the iris. In susceptible individuals, the relatively weak peripheral iris is distended forward, closing the chamber angle and blocking the trabecular meshwork, resulting in an increased IOP. Moreover, anatomical abnormalities in PACG include relatively large and thick lenses [2,9]. In this regard, lens factors become more prominent with aging.

According to the 2019 expert consensus on the diagnosis and treatment of PACG [3], phacoemulsification and intraocular lens implantation are recommended as the first-line treatments, and GSL is performed under a gonioscope. Crucially, IOP must be monitored postoperatively. In this study, patients with angle-closure glaucoma complicated by cataracts were sequentially divided into PEI-GSL and PEI groups to undergo surgery. Visual acuity was improved in both groups postoperatively. However, no significant differences between the two groups were observed in IOP reduction at 1 week, 1 month, or 3 months postoperatively. Notably, IOP reduction was more effective at 6 months postoperatively in the PEI-GSL group than in the PEI group. Several factors may account for these findings. Removing the lens and replacing the turbid lens with an intraocular lens with a thickness of less than 1 mm may relieve the pupillary block factor, and the anterior chamber may be deepened [10]. In phacoemulsification, the flushing effect of the anterior chamber perfusion fluid may wash away glycosaminoglycans, increase aperture, induce cell division, promote phagocytosis, and increase permeability in the trabecular meshwork, thus increasing aqueous discharge functioning [11].

Moreover, the surgery-induced release of endogenous prostanoid E2 may expand the outflow pathway of aqueous humor from uveal-sclera and promote aqueous humor outflow. At the same time, contraction of the capsular pouch after surgery may cause the suspension ligament to increase ciliary body traction and enlarge the ciliary body space, which may ultimately expand the uveoscleral pathway and promote aqueous humor outflow [12]. In the PEI-GSL group, sodium hyaluronate was used to create an obtuse anterior chamber angle and compress the adhered anterior chamber angle at the iris root such that the anterior chamber angle was fully opened, thus releasing the anterior chamber angle adhesion closure. Indeed, IOP reduction was more effective in the PEI-GSL group than in the PEI group. Comparing corneal endothelial counts of the two groups at different postoperative time points revealed that GSL did not increase damage to the corneal endothelium, indicating that phacoemulsification combined with GSL is safe. In line with our findings, White et al. [13] reported that phacoemulsification combined with GSL reduced IOP and the need for IOP-lowering drugs and surgery; in addition, Liu et al. [14] reported that phacoemulsification combined with GSL was the best surgical method for the treatment of angle-closure glaucoma with cataracts. The IOP-lowering ability of this operation was comparable to that of phacoemulsification combined with trabeculectomy.

After phacoemulsification and intraocular lens implantation, the stability of the intraocular lens in the pouch determines the stability of CACD. In contrast, the stability of intraocular lens position after surgery may be affected by several factors [15], including the characteristics and shape of capsulorhexis, intraocular lens design and materials, the tension of the suspensory ligament of the lens, vitreous volume and viscosity, and surgical-related factors, such as surgical injury and postoperative inflammatory reactions. Our analysis revealed that the anterior chamber depth tended to be stable from 3 to 6 months postoperatively in both groups. For patients with angle-closure glaucoma, factors such as whether the anterior chamber angle is open, the size of the angle adhesion area, and the severity of the adhesion directly affect aqueous humor outflow. Crucially, we observed that angle width was significantly greater in the PEI-GSL group than in the PEI group at different postoperative time points. IOP-lowering effects were more significant in the PEI-GSL group than in the PEI group at 6 months postoperatively. Based on these findings, we conjecture that phacoemulsification and intraocular lens implantation combined with GSL may reduce the scope of angle adhesion and reopen the adhered anterior angle in patients with angle-closure glaucoma and cataracts, thus achieving the effects of strengthening drainage and lowering IOP. Consistently, Tian et al. [16] reported that GSL surgery reduced the degree of angle adhesion, the impact of acute angle-closure glaucoma was better than that of CACG, and CACG was more likely to result in the reappearance of angle adhesion. In our study, there were 15 patients with CACG. We did not test further comparisons of the differences in surgical effects between acute angle-closure glaucoma and CACG owing to the limited number of participants. Moreover, Wei et al. [17] demonstrated that, for any degree of peripheral anterior chamber angle adhesion before surgery, PEI-GSL could effectively control IOP, and the extent of postoperative peripheral anterior chamber angle re-adhesion was positively correlated with intraoperative mechanical separation. Therefore, we suggest early screening, diagnosis, and treatment for ophthalmic patients with angle-closure glaucoma involving anterior segment anatomy [18].

The increase in IOP in angle-closure glaucoma is caused by static anatomical single factors as well as dynamic characteristics of the eye. Dynamic changes in the iris, ciliary body, choroid, and lens may induce an acute attack of PACG. Quigley et al. [19] proposed that choroidal expansion leads to the anterior displacement of the lens and a reduction in anterior chamber volume, which leads to the formation of pupillary block and a resultant increase in IOP. We previously demonstrated that the CT of patients with primary angle closure was greater than that of healthy individuals, and LPI reduced the CT of patients with primary angle closure [20]. Studies comparing CT between eyes with acute-onset PACG, healthy eyes, and eyes with primary open-angle glaucoma have shown that CT was the greatest in eyes with acute-onset PACG [21,22]. Singh et al. [23] noted that a thicker choroid in PACG eyes might be a risk factor for poor IOP under drug control, and it was a poorer prognostic marker than the contralateral eye. Li et al. [24] found that the CT of patients with PACG was greater than that of healthy individuals. CT did not differ significantly between patients with moderate and severe glaucoma, suggesting that CT is not correlated with glaucoma progression. We compared preoperative and postoperative CT of PACG patients with cataracts and observed that preoperative CT was greater than that at different postoperative time points.

Further, there were no significant differences in CT at 1 week, 1 month, or 3 months postoperatively. The change of CT is dynamic, and the CT we obtained was the thickness of a single node; additionally, the evolution of IOP is also dynamic, and the IOP we measured was the IOP at a particular time point. The CT of PACG with cataracts before surgery was greater than that after surgery, and so was the change in IOP. However, our previous findings showed no correlation between CT and IOP [20].

Moreover, Zhang et al. [25] reported that, owing to the limitations of the current diagnostic techniques, true changes in CT in patients with acute-onset glaucoma could not be determined accurately, thus precluding the determination of whether choroidal thickening is a risk factor for PACG. PACG may be a group of diseases with similar manifestations but diverse pathogenesis. Changes in choroidal thickening and swelling are characterized by the filling of choroidal vessels and increases in choroidal extravascular volume and choroidal leakage [26]. In our study, the main reason for the difference in CT between the PEI-GSL and PEI groups before and after surgery may be the small sample size included in the study, and the abnormal data from each group may have affected the analysis results.

Our study has some limitations. First, the sample size was small, which might have limited our ability to draw definitive conclusions. Second, some patients were not regularly followed up with after surgery, resulting in the loss of some data, which affected data collection and the overall analysis of the results. Further, longer follow-up studies are needed to determine the effect of GSL on the width of the anterior chamber angle.

## 5. Conclusions

Our findings suggest that patients with PACG who underwent PEI with or without GSL had reduced IOP and CT after surgery. However, a direct causal relationship between choroidal thickening and an increase in IOP remains to be demonstrated. Further studies with larger sample sizes are needed to clarify the relationship between choroidal thickening and glaucoma onset.

## Figures and Tables

**Table 1 jcm-12-00406-t001:** Basic characteristics of patients.

Variable	Mean ± SD	Range/%
Sex		
Male, *n*	7	12.28%
Female, *n*	50	87.72%
Age (years)	69.90 ± 7.57	53–88
PEI-GSL/PEI		
Male, *n*	5/2	71.43%/28.57%
Female, *n*	22/28	44%/56%
Age (years)	(71.11 ± 8.68)/(68.43 ± 6.29)	53–88/57–85
AACG	19/23	45.24%/54.76%
CACG	8/7	53.33%/46.67%

SD, standard deviation; PEI-GSL, phacoemulsification and intraocular lens implantation combined with goniosynechialysis; PEI, laser peripheral iridotomy followed by phacoemulsification and intraocular lens implantation; AACG, acute angle-closure glaucoma; CACG, chronic angle-closure glaucoma.

**Table 2 jcm-12-00406-t002:** Comparisons of BCVA and IOP between PEI-GSL and PEI groups.

	BCVA				IOP (mmHg)			
	PEI-GSL	PEI	t	*p*	PEI-GSL	PEI	t	*p*
Preoperative	4.17 ± 0.53	4.51 ± 0.29	−3.007	0.005 **	20.54 ± 11.78	19.24 ± 4.76	0.533	0.598
1 week after surgery	4.61 ± 0.41	4.80 ± 0.15	−2.271	0.030 *	15.68 ± 3.97	14.73 ± 3.93	0.908	0.368
1 month after surgery	4.64 ± 0.35	4.81 ± 0.14	−2.343	0.025 *	13.86 ± 2.61	14.46 ± 3.89	−0.699	0.488
3 months after surgery	4.64 ± 0.34	4.80 ± 0.14	−2.394	0.022 *	14.27 ± 1.96	15.29 ± 3.10	−1.501	0.140
6 months after surgery	4.66 ± 0.28	4.82 ± 0.12	−2.616	0.013 *	13.70 ± 2.25	15.70 ± 3.32	−2.629	0.011 *
F	36.837	26.534			4.580	10.075		
*p*-value	<0.001 **	<0.001 **			0.007 *	<0.001 **		

* *p* < 0.05; ** *p* < 0.01. PEI-GSL, phacoemulsification and intraocular lens implantation combined with goniosynechialysis; PEI, laser peripheral iridotomy followed by phacoemulsification and intraocular lens implantation; BCVA, best corrected visual acuity; IOP, intraocular pressure.

**Table 3 jcm-12-00406-t003:** Comparisons of corneal endothelial counts between PEI-GSL and PEI groups.

	Corneal Endothelial Counts (/mm^2^)			
	PEI-GSL	PEI	t	*p*
Preoperative	2602.26 ± 359.01	2600.10 ± 369.77	0.022	0.982
1 week after surgery	2414.15 ± 378.64	2418.73 ± 361.86	−0.047	0.963
1 month after surgery	2447.07 ± 372.13	2411.30 ± 436.14	0.331	0.742
3 months after surgery	2460.00 ± 362.94	2434.73 ± 395.25	0.250	0.804
6 months after surgery	2475.26 ± 382.29	2491.17 ± 363.52	−0.161	0.873
F	12.903	17.011		
*p*-value	<0.001 **	<0.001 **		

** *p* < 0.01. PEI-GSL, phacoemulsification and intraocular lens implantation combined with goniosynechialysis; PEI, laser peripheral iridotomy followed by phacoemulsification and intraocular lens implantation.

**Table 4 jcm-12-00406-t004:** Comparisons of CACD between PEI-GSL and PEI groups.

	CACD (mm)			
	PEI-GSL	PEI	t	*p*
Preoperative	2.08 ± 0.35	2.07 ± 0.21	0.137	0.891
1 month after surgery	3.01 ± 0.36	3.04 ± 0.40	−0.384	0.703
3 months after surgery	3.19 ± 0.25	3.18 ± 0.35	0.130	0.897
6 months after surgery	3.26 ± 0.24	3.22 ± 0.37	0.472	0.639
F	123.762	120.247		
*p*-value	<0.001 **	<0.001 **		

** *p* < 0.01. PEI-GSL, phacoemulsification and intraocular lens implantation combined with goniosynechialysis; PEI, laser peripheral iridotomy followed by phacoemulsification and intraocular lens implantation; CACD, central anterior chamber depth.

**Table 5 jcm-12-00406-t005:** Comparisons of AA between PEI-GSL and PEI groups.

		PEI-GSL	PEI	t	*p*
AA12	Preoperative	4.30 ± 7.08	8.17 ± 9.82	−1.721	0.091
	1 month after surgery	21.64 ± 5.84	13.60 ± 10.11	3.722	0.001 **
	3 months after surgery	21.43 ± 5.53	14.55 ± 9.87	3.286	0.002 **
	6 months after surgery	19.67 ± 5.90	14.76 ± 10.60	2.188	0.034 *
AA3	Preoperative	11.92 ± 8.52	13.70 ± 9.41	−0.747	0.458
	1 month after surgery	25.29 ± 7.96	18.50 ± 11.05	2.635	0.011 *
	3 months after surgery	24.18 ± 8.05	17.12 ± 9.37	3.035	0.004 **
	6 months after surgery	24.58 ± 7.52	16.36 ± 8.91	3.740	<0.001 **
AA6	Preoperative	6.95 ± 8.52	12.77 ± 10.12	−2.335	0.023 *
	1 month after surgery	21.14 ± 8.00	16.52 ± 7.35	2.275	0.027 *
	3 months after surgery	20.65 ± 8.43	16.27 ± 6.70	2.181	0.033 *
	6 months after surgery	19.15 ± 7.84	15.40 ± 6.60	1.957	0.055
AA9	Preoperative	9.87 ± 9.63	14.41 ± 9.79	−1.762	0.084
	1 month after surgery	25.58 ± 8.98	18.71 ± 9.53	2.794	0.007 **
	3 months after surgery	24.03 ± 9.36	17.87 ± 9.36	2.481	0.016 *
	6 months after surgery	22.60 ± 8.38	16.66 ± 8.29	2.688	0.010 *

* *p* < 0.05; ** *p* < 0.01. PEI-GSL, phacoemulsification and intraocular lens implantation combined with goniosynechialysis; PEI, laser peripheral iridotomy followed by phacoemulsification and intraocular lens implantation; AA, open angle.

**Table 6 jcm-12-00406-t006:** Comparisons of preoperative CT between PEI-GSL and PEI groups.

	N3	N2	N1	SF	T1	T2	T3
PEI-GSL	207 ± 45	242 ± 44	279 ± 42	327 ± 56	302 ± 51	283 ± 51	257 ± 50
PEI	220 ± 24	262 ± 22	297 ± 14	357 ± 25	332 ± 22	308 ± 16	277 ± 19
t	−1.383	−2.121	−2.177	−2.644	−2.854	−2.370	−1.948
*p*-value	0.175	0.041 *	0.037 *	0.012 *	0.007 **	0.024 *	0.060

* *p* < 0.05; ** *p* < 0.01. PEI-GSL, phacoemulsification and intraocular lens implantation combined with goniosynechialysis; PEI, laser peripheral iridotomy followed by phacoemulsification and intraocular lens implantation.

**Table 7 jcm-12-00406-t007:** Comparisons of CT pre- and postoperatively in the PEI-GSL group.

	N3	N2	N1	SF	T1	T2	T3
Preoperative	207 ± 45	242 ± 44	279 ± 42	327 ± 56	302 ± 51	283 ± 51	257 ± 50
1 week after surgery	202 ± 43	236 ± 44	270 ± 40	319 ± 54	294 ± 51	275 ± 48	252 ± 49
1 month after surgery	201 ± 43	236 ± 44	269 ± 39	318 ± 54	294 ± 51	274 ± 48	251 ± 49
3 months after surgery	202 ± 43	236 ± 44	270 ± 40	319 ± 54	294 ± 51	274 ± 47	251 ± 49
6 months after surgery	198 ± 44	233 ± 43	267 ± 40	314 ± 54	289 ± 49	268 ± 46	243 ± 52
F	22.967	10.343	19.683	27.553	17.455	12.937	11.884
*p*-value	<0.001 **	<0.001 **	<0.001 **	<0.001 **	<0.001 **	<0.001 **	<0.001 **

** *p* < 0.01. PEI-GSL, phacoemulsification and intraocular lens implantation combined with goniosynechialysis.

**Table 8 jcm-12-00406-t008:** Comparisons of CT pre- and postoperatively in the PEI group.

	N3	N2	N1	SF	T1	T2	T3
Preoperative	220 ± 24	262 ± 22	297 ± 14	357 ± 25	332 ± 22	308 ± 16	277 ± 19
1 week after surgery	215 ± 25	256 ± 21	289 ± 16	349 ± 24	324 ± 24	300 ± 17	273 ± 19
1 month after surgery	215 ± 25	255 ± 21	288 ± 16	348 ± 23	324 ± 24	300 ± 17	273 ± 20
3 months after surgery	215 ± 25	256 ± 22	289 ± 17	349 ± 24	324 ± 24	300 ± 17	271 ± 20
6 months after surgery	212 ± 24	253 ± 21	286 ± 16	346 ± 24	319 ± 23	294 ± 16	269 ± 19
F	17.631	33.808	24.264	15.522	28.404	24.781	7.370
*p*-value	<0.001 **	<0.001 **	<0.001 **	<0.001 **	<0.001 **	<0.001 **	<0.001 **

** *p* < 0.01. PEI, laser peripheral iridotomy followed by phacoemulsification and intraocular lens implantation.

## Data Availability

The data presented in this study are available upon request from the corresponding author.

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
