# Peer review of "Changes in Anterior Chamber Angle and Choroidal Thickness in Patients with Primary Angle-Closure Glaucoma after Phaco-Goniosynechialysis"

_jcm, 2023, doi:10.3390/jcm12020406_

Round 1

Reviewer 1 Report

This paper describes an investigation on the changes in angle width and choroidal thickness (CT) before and after phacoemulsification intraocular (PEI) lens implantation combined with goniosynechialysis (GSL) in patients with primary angle-closure glaucoma (PACG) complicated by cataracts. The study is designed as a prospective cohort study on 60 patients with PACG complicated by cataracts at the Department of Ophthalmology of Shaoxing People’s Hospital, China. Patients underwent phacoemulsification intraocular lens implantation combined with GSL (PEI-GSL group) or laser peripheral iridotomy followed by phacoemulsification intraocular lens implantation (PEI group). The authors got a clever way of assessing difference between the two sample populations by measuring several eye parameters before surgery and after surgery with a regular follow-up schedule. Statistically significant differences could be found for several parameters. Main outcome of the study is the suggestion of a relationship between CT and angle-closure glaucoma onset in PACG patients with cataracts, never established before. The authors, due to the small patient sample size available to them, were unable to establish a direct causal relationship between choroidal thickening and an increase in intraocular pressure.

The topics are clearly introduced, but the paper refers only to China, the Chinese situation, and Chinese results: a few good and state-of-the-art international references should be added. The materials and the methods are carefully and thoroughly described. The results are given and thoroughly discussed with an interesting discussion of several pertinent aspects, reaching pertinent conclusions. The overall contents and methods are sound. The limitations of the study are clearly presented, and the authors stressed the need of further studies with a larger sample size to clarify the relationship between choroidal thickening and glaucoma onset. This paper is the result of hard and good work. My congratulations to the authors.

I recommend some minor changes and modifications to improve the manuscript. Individual points are discussed below.

Individual points to be considered:

Abstract, line 9: please replace “to observe” with "at observing". 

Abstract, line 10: please add the acronym “(PEI)” after "phacoemulsification intraocular".

Abstract, line 11: here and all throughout the paper please replace “complicated with” with "complicated by". 

Abstract, line 13: please add “, China.” after "Hospital".

Abstract, line 14: please add the acronym “(LPI)” after "laser peripheral iridotomy”. 

Chapter “Introduction”, line 29-33: the introduction is too focused on China. Just less references to China should be given and more international references added. This is true all throughout the text and should be duly taken into account also since the authors this way overlook recent results obtained by foreign specialist.

Chapter “Introduction”, line 31: please remove "in China". 

Chapter "Introduction" line 33: in this respect, besides the quoted references, the authors should also consider the results from this recently published paper:

“I. RIVA, E. MICHELETTI, F. ODDONE, C. BRUTTINI, S. MONTESCANI, G. DE ANGELIS, L. ROVATI, R.N. WEINREB, L. QUARANTA, 2020. “Anterior Chamber Angle Assessment Techniques: A Review”, J. Clin. Med., 9, 12, #3814.”

Chapter “Introduction”, line 41-42: the authors correctly state that " Indeed, there is a paucity of studies in this area". Please say more about it and/or add references. 

Chapter “Introduction”, line 42: please replace “to explore” with "at exploring".

Chapter “Introduction”, line 46-47: please replace “Further” with "Furthermore". 

Chapter "Materials and Methods", line 59: please explain the sentence “Inclusion criteria were the diagnosis of PACG and cataract requirements”, not so clear in the present shape.

Chapter "Materials and Methods", line 73: to avoid duplications, please replace "in all patients with PACG with cataracts" with "for all PACG patients with cataracts".

Chapter "Materials and Methods", line 87: please remove "of the patient".

 Chapter "Materials and Methods", line 121: please clarify "the macular".

 Chapter "Materials and Methods", line 122: please replace "comprrised " with "included". 

Chapter "Materials and Methods", line 154: please replace "HuynhFeldt” with ”Huynh-Feldt”.

Chapter "Materials and Methods", line 159: please replace "Levene’s” with ”Levene”.

Chapter "Results ", line 169: please replace "their regular" with "for them sequential".

Chapter "Results ", line 246-247: please clarify "revealed a correlation between CT at different time points”. Please explain with what is the correlation between CT.

Chapter "Results ", line 260-261: the same as the previous point. Please clarify "revealed a correlation between CT at different time points”. Please explain with what is the correlation between CT.

Chapter "Discussion" line 274-275: not only in China, but everywhere !!! Please take this fact into proper account. The authors should add also non-Chinese results to their quite pertinent discussion. Moreover, non-Chinese references should be also added. 

Chapter "References": in some references were originally in the Chinese language, this should be mentioned for each individual reference.

Reviewer 2 Report

The manuscript reported “The changes in anterior chamber angle and choroidal thickness in patients with primary angle-closure glaucoma after phaco-goniosynechialysis”. The manuscript is well written and highly informative. There are some comments as following.

Title

The manuscript seems to study on the effects of phacoemulsification with intraocular lens implantation comparing between with and without goniosynechialysis in patients with PAC/PACG. These effects included not only the changes of anterior chamber angle and choroidal thickness but also IOP, endothelial cell count, angle opening, CACD etc. The comparison between groups had been done significantly, except choroidal thickness. The title may not completely relevant to the information discussed in the manuscript.

Abstract

The abstract is brief but lack of the brief conclusion of other related finding such as “Post-operatively, the IOP, the endothelial cell count were significantly decreased and the BCVA, CACD were significantly increased in both groups but not significantly different between groups.”

The authors could not conclude that “the present study suggests a relationship between CT and angle-closure glaucoma onset in PACG patients with cataracts” because the study did not compare the results with patients without PAC/PACG.

Introduction

Line 29. Glaucoma is the second most common cause blindness of the world and is the most common cause of irreversible blindness.

Line 42. This study could not explore “the role of CT in the pathogenesis and early diagnosis of PACG” but the change of CT after cataract surgery in PAC/PACG with and without GSL. Furthermore, there was no comparison of post-operative CT between group so the author could not mention the outcome “compared the change of CT between groups”.

Materials and methods

There are several ways to perform goniosynechialysis. The surgical technique of goniosynechialysis in this study was visco-goiosynechialysis. The term “goniosynechialysis” usually refers to the mechanical separation of the peripheral anterior synechiae from the anterior chamber angle with an instrument through gonio-surgery lens. The author should specify the term visco-gonionechialysis throughout the manuscript as well as the title.

The author should clarify the terminology of primary angle-closure suspected (PACS), primary angle-closure (PAC), and primary angle-closure glaucoma (PACG), which is commonly used. If there is no evidence of glaucomatous optic neuropathy or glaucomatous visual field defect, the term glaucoma should be avoided. Several studies have shown the benefits of goniosynechialysis after acute PAC, not chronic PACG. In this study, most of the participants were patients with acute primary angle-closure (acute PAC) (not yet glaucoma). There were 15 chronic PACG enrolled, which may present the different outcomes. The authors may need to discuss or mention this in details.

Results

3.4 Choroidal thickness (CT)

The preoperative CT in PEI-GSL was significantly thinner than PEI groups. There was no comparison of post-operative CT between groups which was a critical point of this study. Due to the unequal preoperative CT, the author should analyze the change of CT after each procedure and perform the comparison of the change of CT after PEI. 

The details of each data collections have been presented informatively in the table. The author should avoid describe the same information in the text. 

Discussion

The author mainly discussed on the comparison of their finding between PEI-GSL and PEI groups but not post-operative CT. Due to the main outcome measures were the change of anterior chamber angle and CT, the author should add the discussion on the difference of CT after PEI-GSL and PEI-PI.  

Conclusion

The authors should amend the conclusion to be relevant to the study. The change of CT in this study could not refer to the relationship or pathogensis of CT and angle-closure. They can conclude only that after PEI with and without goniosynechialysis, the CT was thinner, which may benefit to PACG progression according to the previous hypothesis.
